# Potential Forage Hybrid Markets for Enhancing Sustainability and Food Security in East Africa

**DOI:** 10.3390/foods12081607

**Published:** 2023-04-10

**Authors:** John Jairo Junca Paredes, Jesús Fernando Florez, Karen Johanna Enciso Valencia, Luís Miguel Hernández Mahecha, Natalia Triana Ángel, Stefan Burkart

**Affiliations:** CIAT (the International Center for Tropical Agriculture), Crops for Nutrition and Health, Tropical Forages Program, Palmira 763537, Colombia

**Keywords:** potential markets, improved forages, food systems, food security, forage hybrids, nutrition

## Abstract

The cattle sector is strategic for both the economic development and food security of Africa, but the low availability and quality of forage puts the most vulnerable population at risk. Hybrid forages are an alternative for enhancing both food security and sustainability of the sector but adoption levels are still low in Africa, which is related to various factors such as the availability of seeds. This document analyzes potential markets for new interspecific hybrids of *Urochloa* and potential hybrids of *Megathyrsus maximus*, adapted to the environmental conditions of eastern and partially western Africa, applying a four-step methodology based on estimating (i) required forage amounts for each country according to its dairy herd, (ii) potential hectares for forage cultivation based on (i), (iii) hectares that can be covered by the two hybrids of interest according to a Target Population of Environment approach, and (iv) potential market values for each country and hybrid. The results show a potential market of 414,388 ha for new interspecific hybrids of *Urochloa* and 528,409 ha for potential hybrids of *Megathyrsus maximus*, with approximate annual values of 73.5 and 101.1 million dollars, respectively. Ethiopia, Tanzania, and Kenya hold a market share of 70% for *Urochloa*, and South Sudan, Ethiopia, and Tanzania a 67% market share for *Megathyrsus maximus*. The results will help different actors in decision-making, i.e., regarding private sector investments in forage seed commercialization or public sector incentives supporting adoption processes, and thus contribute to increasing food security and sustainability in the region.

## 1. Introduction

The cattle sector in East Africa is strategic in the fight against hunger and poverty. It provides employment and, at least partially, the livelihood for about 70% of the rural inhabitants in the dry areas of West and East Africa, i.e., for about 110 to 120 million people [1,2,3]. Dryland cattle farmers have on average 1.2–2 Tropical Livestock Units (TLU) per capita, which makes them vulnerable to deteriorations in their living conditions in the face of droughts, disease outbreaks, or any other type of unforeseen event, since it is estimated that they require around 3–4 TLU to stay above the poverty line [1]. As a livelihood, the sale of milk is the predominant way of obtaining benefits from cattle farming as it generates income for covering the daily expenses of families (e.g., for food, medicine, clothes, or schooling) [2,4,5,6,7,8,9,10,11]. Culling is secondary, and cattle are used as savings, a store of value that generates income in the short-term and that is saved for difficult periods (e.g., for the purchase of feed) or important expenses such as schooling, converting it into both a means of savings and insurance [4,5,6,12,13,14,15,16,17]. Regarding the production system, average farms in the region have less area than necessary to maintain one cow and her calf, in such a way that the predominant practice is cut-and-carry of forages for cattle feeding in stables [18].

In the region, there exists both scarcity and low quality of forage, a situation which is accentuated in dry seasons and influenced by climate change [2,19,20,21]. This, combined with a lack of quality, efficient, and sustainable production is manifested in poor supply levels of animal sourced foods and has affected food security over time [22]. Between March and July 2022, in Kenya, Somalia, and Ethiopia, the number of children affected by acute hunger, malnutrition, and thirst increased from 7.25 to almost 10 million [22]. Droughts and price hikes over the recent months, related to the COVID-19 pandemic and the war in Ukraine among others, worsen a problem that has historically marked the region [22].

Against this background, a technological change in food systems is needed to overcome the problems of food insecurity and malnutrition. The transition from traditional or low-productive pastures to sustainable forage-based cattle systems with high performance and nutritional quality is one solution that can help the situation [18,21,23,24,25]. The adoption of improved forage materials by cattle producers allows for obtaining quality animal feed and thus food of animal origin in higher quantities and quality. In an environment where poverty and famines are common, new forage technologies offer the opportunity to provide quality meat and milk to the most vulnerable population. In Kenya, for example, efforts are being made among public, private, and research actors to strengthen both the commercialization and adoption of higher quality forage grass seeds and splits, such as *Urochloa* (syn. *Brachiaria*) and *Megathyrsus maximus* (syn. *Panicum maximum*) hybrids and varieties, among dairy farmers [26].

Before improved forages can be adopted and disseminated; however, they need to be developed and this development needs to be adjusted to the region’s needs. Regarding forage technology development, forage breeding is among the most promising alternatives for East Africa [21,25]. Over the last century, plant breeding has contributed significantly to raising crop yields [27]. The improved forages and forage hybrids developed e.g., by the forage breeding programs of the International Center for Tropical Agriculture (CIAT) or the Brazilian Agricultural Research Corporation (EMBRAPA), can increase both the productivity and quality of feed for the dairy sector, and thus contribute to improving food security, incomes, and livelihoods of dairy producer families [19,25]. Likewise, the adoption of improved forage technologies generates positive environmental externalities, for example a reduction of greenhouse gas emissions from cattle systems [21,25,28]. Breeding research for the development of future forage hybrids for the region emphasizes on traits such as higher nutritional quality, nitrogen-use-efficiency (NUE), and the ability to regenerate and avoid soil degradation [29]. Despite both the economic and environmental benefits of forage hybrids, the adoption process of existing commercial hybrids in East Africa is slow and accompanied by numerous challenges, including a lack of awareness and knowledge of the technologies, low state investment, poorly developed input and output markets [28], and, above all, a poorly developed forage seed system and market [30,31,32,33].

Under this scenario, the hypothesis of this research article is that an important potential market exists for new forage hybrids in East Africa that can be captured by the private forage seed sector and contribute to increasing livelihoods, food security, and nutrition in the region. This market basically emerges from the huge potential for the adoption of hybrid forages resulting from the superior characteristics the materials offer regarding productivity, adaptability to the environment, and nutritional quality that improve both animal productivity and livelihoods of dairy farmers, while contributing to food security and environmental sustainability in the region. The mentioned hypothesis leads to the following research questions: (i) How big is this potential market for new forage hybrids of *Urochloa* and *Megathyrsus maximus* in East Africa, and (ii) what needs to be done so that the potential can be captured and adoption will happen? The objective of this article is thus to provide an analysis of this potential market for new interspecific *Urochloa* and *Megathyrsus maximus* hybrids for East Africa, and some West African countries. Particularly, this study aims at (i) providing a market segmentation for the two hybrids of interest, (ii) estimating the area that can be covered by the two hybrids in each country of analysis (size of potential market), (iii) estimating the commercial annual and total values the two hybrids could generate in the region (market value), and (iv) describe how this market can be captured to make large-scale adoption reality.

This study is a contribution to the efforts made by the CGIAR Initiative on Market Intelligence, which seeks to expand the social impact of technologies developed by the CGIAR centers in areas such as nutrition, gender equality, and climate change [2,34]. The study thus works towards various of the Sustainable Development Goals, namely no poverty (SDG-1), zero hunger (SDG-2), and climate action (SDG-13), among others. The forage hybrid market segments used in this study were previously identified within the CGIAR EiB (Excellence in Breeding) and BPAT (Breeding Program Assessment Tool) programs between 2017 and 2019. EiB aims at modernizing crop improvement programs for better tackling the needs of farmers from low- and middle-income countries [35]. The BPAT tool is applied for the revision of the different components, capacities, and technical aspects of existing breeding programs, aimed at enhancing the rates of genetic gain [36]. This study is an important contribution to both scientific literature and the development of the forage seed sector in the region since market studies on improved forages are extremely scarce. The results provide relevant information to better understand the possibilities and economic opportunities that a massification of new forage hybrids could have. Similarly, market segmentation allows reducing the levels of uncertainty in terms of where to promote different types of new technologies and thus reduces the risks associated with adoption. This is key since the ultimate objective of plant breeding is to develop new and superior technologies that are adopted by farmers and contribute to changing their livelihoods and the economic development of a country or region.

This study applied quantitative methods for the forage hybrid market segmentation and valuation exercises. First, the forage hybrid market segments were described, based on existing information from the EiB and BPAT exercises and expert consultations. Second, forage requirements for feeding the national cattle herds were determined and the area required to produce the forages were defined based on secondary data from the Food and Agriculture Organization of the United Nations (FAO) and expert consultation. Third, based on existing information from a study that defined potential geographical suitability for the hybrids of interest in the region based on geographic information system (GIS) and multivariate cluster analysis [37], the size of the potential market for the two materials was determined. Fourth, the commercial values of the potential markets were defined using geometric averages for market prices. Lastly, factors that determine the adoption of new forage hybrids were retrieved from literature. The analysis includes several countries in eastern and western Africa, namely Ethiopia, Kenya, South Sudan, Tanzania, Uganda Nigeria, and Mali, which were identified as priority countries in 2020 as part of the EiB and BPAT exercises of CIAT’s Tropical Forages Breeding Programs.

This document is composed of this introduction (Section 1), a brief literature review to shed light on both existing scientific evidence and applied methods (Section 2), a description of the forage market segments of interest to provide the required background information for understanding the technologies (Section 3), a section on materials and methods (Section 4), a combined results and discussion section (Section 5), and a section with the major conclusions (Section 6).

## 2. Brief Literature Review on Similar Studies

This section provides a brief overview on the literature related to market analysis of agricultural technologies and their adoption in agricultural systems with the purpose of providing insights in (i) the current research on the topic, and (ii) the applied methodologies. The methodologies used range from basic survey analysis to discrete choice models impact evaluations, multivariate techniques, and GIS such as the one used in this article.

In Indonesia, in a sample of 182 farmers, the demand for clean potato seeds in formal and informal markets was analyzed, obtaining an estimate of the willingness to pay for a higher quality seed. This study found that potato growers understand the advantages of these seeds and that the main limitation for adoption is the high price of the material. The study estimates that a large number of farmers would obtain benefits from these materials, since yield differences between 30% and 50% were estimated. In some markets, this would increase the willingness to pay by up to 37% [38]. In Kenya, the potential adoption of biotechnologies that protect maize from various types of fungi was analyzed. In a sample of 480 households, a potential adoption of 82% of these technologies was estimated with a logistic discrete choice model, which allowed to infer that both formal education and knowledge of the new technology influence adoption. Likewise, high-income farmers are more willing to make changes to their production systems. The condition of poverty generates significant risk aversion and is a limitation for adoption [39].

Other scholars have used impact evaluation methodologies with the objective of estimating the potential adoption of improved crop technologies, particularly by estimating the average treatment effect (ATE). In this context, the treatment poses scenarios in which the population is exposed to the knowledge of the new technology and can access the product, and thus actual and potential adoption can be obtained. Following this approach, an evaluation in Nigeria estimated the potential adoption of new rice varieties using a probit model. Producers with a higher educational level, age, access to extension services, and knowledge of local varieties were more likely to know and acquire improved seeds. The actual adoption rate was 19%. The results indicate that having the knowledge places the adoption potential at 54% and if producers can obtain the new seed, the value increases to 62% [40]. In Benin, the adoption of improved corn varieties was evaluated in a sample of 490 farmers with a probit model approach. The results revealed that literacy, the relationship with institutions, the area planted with corn, and income from corn production are the main determinants of adoption. A total of 84% of the producers knew the improved seed, with which the adoption was located at 78%. A global knowledge of the technology would imply a potential adoption rate of 93% [41]. A study in Uganda shows the potential adoption of drought-resistant maize in three scenarios, based on three probit models for the evaluation, referring to (a) the producer’s knowledge of the new technology, (b) producer knowledge and availability of planting material, and (c) producer knowledge, availability of planting material, and affordable market prices. Based on this, an actual adoption rate of 14% was estimated. The potential adoption rates for the three scenarios were 22%, 30%, and 47%, respectively [42]. In Mali, potential adoption rates for eight climate-smart agriculture technologies were estimated with a logit model. Among these are varieties of crops resistant to droughts, organic fertilizer, and agroforestry. With a sample of 300 families, the observed adoption was between 39% and 77%, depending on the technology. In terms of access to knowledge, potential adoption fluctuated between 55% and 81%. Among the factors that influence adoption, the number of farm workers, access to subsidies, and capacity building/training were identified [43]. The reviewed studies provide consensus that the adoption of new technologies depends on the dissemination of both the technology itself and knowledge about it.

Another method of evaluating the potential market for new improved seed technologies are GIS and multivariate statistical analyses. These techniques use environmental data from official statistics and meteorological sources. For example, using a basin-level hydrological model and simulation techniques, the areas with the best yield of both total aerial biomass and cocoa beans in the humid tropics of southeastern Mexico were identified. The objective was to identify the areas with the greatest potential for productivity and economic benefit. The results show that cocoa is profitable when more than 770 kg of grain/ha is produced and that there are 223,000 ha with potential for this crop. The study uses information on climate, hydrology, soil, plant growth, other environmental variables, and management practices [44]. In Nuevo León, Mexico, the areas with the best productive possibilities for 16 crops were identified using thermal data, soil type data, and thematic maps. Results show that basic grains, vegetables, and fruit trees are suitable in more than 50% of the region’s agricultural area. Another relevant result is the more precise identification of regions with frost phenomena. The study highlights the importance of this type of analysis to reduce the risk involved in any business activity in the agricultural sector [45]. Moreover, in Mexico, a potential market index at the state level was developed for corn by building the indicator with variables such as the area planted with traditional and improved varieties. This information was combined with socioeconomic data to obtain the areas with the greatest potential for adoption of improved materials. The regions with the best prospects are the Lower Pacific Tropics with 1,485,272, Valles Altos with 954,197, and the Humid Tropics with 534,279 bags of seed [46].

In the case of forages, market studies are scarce but literature provides insights about a series of difficulties related to technological change. The final adoption decision is in the hands of the producers and this in turn depends on various elements [47]. Institutional, logistical, infrastructural, and information factors are important constraints for adoption. In East Africa, these bottlenecks have been identified through qualitative, quantitative, and mixed methods. In Tanzania, the climate, insufficient availability of seeds, technical deficiencies, low productivity of local livestock, low milk prices, and few incentives for labor in dry seasons are the determinants of low adoption rates of improved forages [48,49]. In Ethiopia, forage adoption is affected by poor transport infrastructure, which increases production costs. Similarly, logistical difficulties affect the distribution of surplus milk. These elements end up offsetting the productivity and profitability gains obtained with the adoption of improved forages [50,51]. Moreover, in Ethiopia, political factors, such as high staff turnover in public institutions, affect the dynamics of the forage sector and create scenarios of uncertainty [51]. In Kenya, households with no land ownership, low educational level of the head of household, large families, and far away from markets are less likely to adopt forage technologies [52]. In Malawi, dairy processing is operating at 20% capacity and consumption is below the African average. Improved forages would significantly contribute to the development of the sector. However, ignorance of forage technologies, market entry barriers, and inadequate approaches in extension programs slow down the adoption of these materials [53]. Several of the studies agree that extension services are one of the main bottlenecks for the adoption and sustainability of new technologies. Technical support does not usually accompany all production stages, which generates significant losses in the early stages of development [48,49,53,54]. As literature shows, to face these limitations and increase adoption rates, it is necessary to consolidate relations between the public and private sectors and research to raise awareness among the rural population about the advantages of new forage technologies regarding productivity, costs, and sustainability. Likewise, it is necessary to strengthen the access to technical and entrepreneurship training, which allows for long-term sustainability of the new technologies.

## 3. Market Segments for the Forage Hybrids of Interest

This section provides insights into (i) past and current *Urochloa* and *Megathyrsus maximus* breeding efforts and advances at CIAT, (ii) the market segmentation exercise and characteristics for interspecific *Urochloa* hybrids, and (iii) the market segmentation exercise and characteristics for *Megathyrsus maximus* hybrids.

### 3.1. Urochloa and Megathyrsus maximus Breeding at CIAT

Potential markets for new hybrid materials of *Urochloa* and *Megathyrsus maximus* species are analyzed in this study [2,55,56]. Hybrids are the product of genetic improvements and combine the superior traits of different materials. CIAT began this line of research in 1987 with an interspecific breeding program with *U. brizantha* (CIAT-6294 cv. Marandú), *U. decumbens* (CIAT-0606, cv. Basilisk), and *U. ruziziensis* (BR4X-44-2) [2,47,57]. This research, together with the efforts of the private forage seed sector, allowed the formal release of various forage hybrids, including *U.* hybrids cv. Mulato I and II, Cayman, Camello, and Cobra [58,59]. These *Urochloa* hybrids are interspecific, which means that different species of the same genus were crossed to obtain an improved hybrid [2,60]. Mulato I and II were the first forage hybrids launched in Africa in 2005. Much later, Cayman and Cobra followed in 2019, and Camello in 2020 (Papalotla 2022, personal communication). The market does not yet count with hybrids of *Megathyrsus maximus*, but development has started several years ago, and the release of a first hybrid is only a matter of time. Although CIAT’s forage breeding program also focuses on the development of hybrids of *U. humidicola*, they are destined for moist soils [61] and thus, not adapted to the conditions of most regions of East Africa [2].

The predominant characteristics a region has regarding its soils, climate, and agricultural practices are key for identifying forage hybrid markets. The technical information on potential new *Urochloa* and *Megathyrsus maximus* hybrids presented in this section is derived from field measurements in pilot experiments. However, the large number of trials required at the early breeding stages made it unfeasible to conduct the pilots in Africa directly [37]; (V. Castiblanco, personal communication, 13 June 2019). Thus, CIAT’s breeding programs identified areas with similar geographic and environmental characteristics to those of East Africa and applied the initial hybrid trials in Colombia [2,37].

### 3.2. Market Segmentation for Urochloa Interspecific Hybrids

*Urochloa* interspecific hybrids are destined for sub-humid tropical savannahs with low fertility and acid soils in eastern and southern Africa [2]. African soils suffer from desertification, which is negatively affecting yields and undermining the resilience of the agriculture and livestock sector, two detrimental elements of subsistence farming in Africa [62]. *Urochloa* hybrids are used for two purposes, namely (i) free grazing and (ii) cut-and-carry for feeding in stables. Important hybrid traits for the region include performance, response to pests and diseases, targeted production systems, and seed production potential.

The projected performance of new *Urochloa* interspecific hybrids, based on the pilots by CIAT’s breeding program, is described below. New hybrid materials are expected to have seed yields equal or superior to the existing commercial offer, even when extreme environmental conditions predominate (e.g., heat, drought, water-logging, acid soils) (Table 1). Seed production potential is important because it means greater productivity and efficiency, which allows new products to be competitively priced in the market. Likewise, materials are expected to be performing equal or superior regarding NUE, as inputs are scarce in the region and the use of existing resources needs to be optimized under the premise of sustainable development [63]. Regarding the forage quality, the new pilot tested *Urochloa* hybrids that have a crude protein (CP) content ≥ 10.5% and an in vitro digestibility of dry matter (IVDMD) ≥ 62%, both important measures for feed quality [64,65]. Regarding both shade tolerance (important for silvo-pastoral systems) [66] and palatability (important for the selection by the animal) [67], the new hybrids are expected to reach intermediate to high levels (on a scale of 1 to 9). *Rhizoctonia* leaf blight is one of the major diseases for forages in the region, with up to 50% of the planted *Urochloa* affected [68], and new hybrids should rank ≤ 2 on a scale of 1 to 5. Regarding the resistance to insects, the analysis is still in the stage of development of a phenotyping methodology. However, the already existing *Urochloa* hybrids have a good response to the spittlebug complex (Hemiptera: Cercopidae), but less to *Tetranychus urticae* (red spider mite), an insect that has affected the Mulato II hybrid and the Basilisk variety in East Africa, for example in Kenya [2,69].

The production system for which interspecific *Urochloa* hybrids are aimed at is dryland cattle production, where rainfed agriculture is predominant and no artificial irrigation techniques are implemented—which comprises large parts of the African soils [70]. The traits considered essential for new interspecific *Urochloa* hybrids are seed yield, forage CP and IVDMD contents, and resistance to pests and diseases. Competitors of new hybrids currently available on the market are (a) Mulato II, (b) Cayman, (c) Camello, and (d) Cobra [2,59]. Table 1 lists the main characteristics of these competitors.

**Table 1 foods-12-01607-t001:** Potential competitors for new interspecific hybrids of *Urochloa*.

Characteristics	Mulato II	Cayman	Camello	Cobra
Main features	Good response to drought, acid soils, and high temperatures [2] Combines the best features of other hybrids [2]	Tolerant to humidity and waterlogging [59]	Drought tolerance, quick establishment, good for acid soils [59]	High yield, vertical growth that facilitates cutting [59,71]
Resistance to pests and diseases	spittlebug [59]	spittlebug [61]	spittlebug [71]	spittlebug [71]
Required soil fertility level	medium, high [59]	humidity [59]	medium [59]	high (for higher yields) [59]
Palatability	very good [59]	very good [59]	very good [59]	very good [59]
CP (%)	14–22 [59]	10–17 [59]	14–16 [71]	14–16 [71]
IVDMD (%)	55–66 [59]	58–70 [59]	62 [71]	69 [59]
Yield (t/ha/cut)	25 [72]	<24 [73]	27–30 [71]	35–40 [71]
Main use	grazing [59]	grazing [59]	grazing [59]	cut-and-carry [59]

Source: own elaboration based on [2,59,61,71,72,73].

### 3.3. Market Segmentation for Megathyrsus maximus Hybrids

*Megathyrsus maximus* hybrids are destined to cut-and-carry production systems in the sub-humid tropical savannah of eastern and southern Africa, where highly productive and fertile soils predominate. According to the pilot tests carried out in Colombia, *Megathyrsus maximus* hybrids are expected to have seed yields equal or superior to the existing commercial offer, even when extreme environmental conditions predominate (e.g., heat, drought, water-logging, acid soils) and NUE is considered (Table 2). Regarding the forage quality, the hybrids have a crude protein (CP) content ≥ 10.5% and an in vitro digestibility of dry matter (IVDMD) ≥ 62%. In addition, hybrids have a moderate to high Biological Nitrification Inhibition (BNI) potential, reducing the use of fertilizers in feed production and thus generating savings both in production costs and greenhouse gas emissions [2,74].

Similar to the case of interspecific *Urochloa* hybrids, the production system for *Megathyrsus maximus* hybrids is rainfed. Potential competitors already available on the market are (a) *Megathyrsus maximus* cv. Mombasa, (b) *Megathyrsus maximus* cv. Tanzania, (c) *Megathyrsus maximus* cv. Massai, and (d) *Urochloa* hybrid cv. Mavuno [2,61,72,75,76,77]. Table 2 summarizes the main characteristics of these materials.

## 4. Materials and Methods

This section gives an overview on the materials and methods used in this study. First, a brief overview is provided on the information sources consulted for estimating potential forage hybrid markets and market values. Second, the methods for the estimation of both potential markets and market values applied in this study are presented. This includes four main steps, namely (i) estimating the required forage amount for each country, based on the present dairy herd and its needs, (ii) estimating the potential annual hectares for forage cultivation based on this need, (iii) assigning a proportion of the required hectares to the two forage hybrids of interest, based on a Target Population of Environment (TPE) approach, and (iv) estimating the potential market value for each country and forage hybrid of interest. The applied methodological steps are summarized in Figure 1.

### 4.1. Information Sources

Given that dairy production is the most relevant cattle activity in the region, the quantitative approximation of the potential hectares of new hybrid forages is based on the information on cattle heads destined for dairy production. In the FAOSTAT Database (Food and Agriculture Organization Statistics) of the FAO, production, crops, and livestock products were entered, and the information was filtered according to the required criteria [78]. In “element”, animals in production were chosen. In “product”, primary livestock (list) was displayed by choosing the option “raw milk from bovine cattle”. Finally, the year 2020 and the countries of interest were selected. The process produced a data table with the information on the heads of dairy cattle per country. In the calculation, this information will be converted into hectares of forages required to feed these animals.

To define the percentage of adoption of each material, the Target Population of Environments (TPE) study of CIAT’s forage breeding programs was consulted, by which the areas that are most suitable for the evaluated forage materials were identified [37]. Information on the market prices of different forage seeds was obtained from the prices published by different seed distributors on electronic commerce platforms for the second half of 2022 [59]. Table 3 summarizes the different variables used in this study and provides information on the sources of information consulted.

### 4.2. Method for the Estimation of Potential Markets and Market Values

This section describes the methods used to estimate potential markets for new interspecific *Urochloa* and *Megathyrsus maximus* hybrids in East and West Africa, particularly in Ethiopia, Kenya, Tanzania, Uganda, South Sudan, Mali, and Nigeria.

#### 4.2.1. Steps 1 and 2—Estimating Forage Requirements and Potential Hectares for Cultivation

Based on data from the FAO on cattle heads for dairy production for the year 2020, the number of hectares required for forage cultivation were calculated. The estimated area is a conservative assumption since new hybrid forages will have increased performance and require fewer area for the same level of production than the existing commercial offer. The estimation considered the following assumptions based on expert consultation [2]: A daily green matter requirement of 60 kg per animal, a forage adoption rate of 15% per year, and a green matter yield of 60 tons per hectare and year.

By means of Equations (1) and (2), the calculation of the potential hectares was carried out. Equation (1) gives the annual forage requirement in tons. With Equation (2), the hectares of forages required to feed the animals were obtained. The measurement is in green matter. Equations (1)–(4) are based on V. Castiblanco and A. Notenbaert (personal communication, 13 June 2019).
(1)$$RAFT=\frac{RDFK\times 365d\times CRL}{1.000},$$
where RAFT is the annual forage requirement in tons, RDFK is the daily forage requirement in kg, and CRL is the number of dairy cattle heads. Substituting this result in Equation (2), the potential hectares are obtained,
(2)$$HaP=\frac{RAFT\times AR}{Ytha},$$

HaP is the potential forage hectares required for forage cultivation, AR is the adoption rate, and Ytha is the average hybrid forage yield per hectare in tons. For example, in 2020, Kenya had 5,112,340 dairy cattle. According to this and the established assumptions, this leads to the following estimation for the annual forage requirement:(3)$$RAFT=\frac{60kg\times 365d\times 5,112,340}{1.000}=111,960,246\;tons,$$
and for the potential hectares required for hybrid forage cultivation:(4)$$\mathrm{HaP}=\frac{111,960,246t\times 15\%}{60}=279,901\;\mathrm{ha},$$

#### 4.2.2. Step 3—Estimating the Market Size: Assigning a Proportion of the Potential Hectares to the Two Forage Hybrids of Interest

The third step was the estimation of the market size through assigning a proportion of the potential area identified in Section 4.2.1 to each of the two hybrids of interest (interspecific *Urochloa* and *Megathyrsus maximus* hybrids). This proportion was obtained from a TPE study conducted by CIAT in 2019 (V. Castiblanco; A. Notenbaert, personal communication, 13 June 2019) [37]. This study applied both GIS and multivariate cluster analysis, and in this way, areas with similar environmental traits in Africa and Colombia could be identified [37] (see Figure 2). This allowed the pilot experiment referenced in Section 3 to be carried out in Colombia. Likewise, four geographic clusters with similar environmental characteristics were identified [37], considering cattle density [80], soil quality data [81], and different precipitation levels [82,83].

For this analysis, two groups are relevant, namely (a) Cluster 2 (good), colored red on the maps, is characterized by higher precipitation levels and better rainfall distribution throughout the year. It provides the conditions for the adoption of potential *Megathyrsus maximus* hybrids, which have high quality and productivity levels, but require good environmental conditions [2]. Cluster 2 represents 28% of the potential area [37]. Moreover, (b) Cluster 3 (hostile), colored blue on the maps, is characterized by low precipitation levels and poor rainfall distribution throughout the year. It provides the conditions for new interspecific *Urochloa* hybrids, which have a medium to high productivity and are very adaptable to difficult environments [2]. Cluster 3 represents 27% of the potential area [37]. These percentages were applied to the HaP (step 2) to define the size of the potential markets (MS) for both new interspecific *Urochloa* and *Megathyrsus maximus* hybrids. This was done for each of the countries of interest. Equations (5) and (6) allow obtaining the respective values [37]:(5)MSU=HaP×27%
(6)MSM=HaP×28%
where MS*U* and MS*M* are the market sizes for new interspecific *Urochloa* and *Megathyrsus maximus* hybrids in hectares.

#### 4.2.3. Step 4—Estimating the Potential Market Values

Finally, the commercial value of the identified market segments was estimated, consulting average market prices for the materials considered as competitors for new interspecific *Urochloa* and *Megathyrsus maximus* hybrids (see Table 1 and Table 2). Due to the high research and development costs, among others, forage hybrid seeds have a higher market price than other forage varieties. Regarding *Megathyrsus maximus*, there are no hybrid materials available on the market yet; hence, direct price references are missing. To obtain market price estimations for these hybrids, the price difference between interspecific *Urochloa* hybrids and other commercial *Urochloa* varieties was applied to the case of *Megathyrsus maximus*, too. For this, geometric averages were used, as they better capture price dynamics [84]. Equations (7) and (8) allow obtaining the respective values [84]:(7)$$PMU=\sqrt[{n}]{P_{1}\times P_{2}\times \cdot \times P_{n}},$$
(8)$$PMM=h\times \sqrt[{n}]{P_{1}\times P_{2}\times \cdot \times P_{n}},\;\;\;con\cdot h>0$$
where PMU and PMM are the averages market prices of *Urochloa* and *Megathyrsus maximus*, respectively. The term h represents the margin that increases the price to level it to the hybrids, and n corresponds to the number of data used for the calculation.

Finally, the value of each market was expressed according to Equation (9) [79]:(9)Vm=MS×S×P,

The market value (Vm) was calculated for each hybrid, considering the market size in hectares (MS), the sowing rate per ha in kg of hybrid seed (S), and the market price (P) of one kg of seed. Following the literature on the subject, 7 kg of forage hybrid seed is required for each ha [79]. The average market prices for *Urochloa* and *Megathyrsus maximus* varieties were US$ 18.42 and US$ 19.87, per kg of seed, respectively, and for the existing interspecific *Urochloa* hybrids, US$ 25.35. The price premium for *Urochloa* hybrids can thus be estimated to be 37.63%. Applying this price premium to the case of *Megathyrsus maximus* results in a potential market price of US$ 27.34 for a kg of hybrid seed.

## 5. Results and Discussion

This section provides the results and discussion of this study. In particular, the results obtained from the different estimations are presented and then put into context with current literature on forage hybrid markets and adoption.

### 5.1. Forage Requirements to Feed the Cattle Herds and Area Required for Forage Cultivation

Dairy production is the most representative cattle activity in the region of analysis. According to the FAO, the dairy cattle herd in Africa for 2020 reached 66,330,001 heads. The largest inventory is concentrated in East Africa, which holds 34,723,481 dairy cattle. Within this region, the largest herds are found in South Sudan (8,432,559 dairy cattle), followed by Ethiopia (7,556,402), Tanzania (7,116,771), Kenya (5,112,340), and Uganda (4,037,038) [78]. To understand the relative importance of the dairy sector, some figures regarding beef cattle are important to consider, since its participation in the region is lower. Africa counts with a total beef cattle herd of 41,720,252 heads, and in East Africa, there are about 14,527,659 beef cattle. The countries with the highest inventory of beef cattle in the region are Ethiopia (4,086,481 beef cattle), Tanzania (3,554,364), Kenya (1,953,734), Uganda (1,217,247), Zambia (1,065,054), and South Sudan (964,884) [78]. Given the importance of the dairy sector in the region, only the information on dairy cattle was considered to estimate potential hectares of improved forages needed to feed the animals according to the methodology exposed in the previous section. A summary of the RAFT and HaP estimates for the analyzed countries can be found in Table 4.

### 5.2. Size of the Potential Markets

The results of the estimation for the potential market (market size) of new interspecific hybrids of *Urochloa* are provided in Figure 3A. The biggest market can be observed in Ethiopia with a potential of 111,703 ha for interspecific *Urochloa* hybrids, followed by Tanzania and Kenya with 105,204 and 75,573 ha, respectively. Uganda and Nigeria are in the middle range with 59,678 and 32,726 ha, respectively, and Mali, another country from West Africa, holds the smallest market potential (29,505 ha). These figures imply that the mere participation of Ethiopia, Tanzania, and Kenya represents about 70% of the potential area of adoption of new interspecific *Urochloa* hybrids in the analyzed countries, and 83% when only the East African countries are considered.

Figure 3B shows the results for the potential market (market size) of *Megathyrsus maximus* hybrids. South Sudan holds the biggest market potential with 129,271 ha, followed by Ethiopia (115,840 ha), Tanzania (109,100 ha), Kenya (78,372 ha), and Uganda (61,888 ha). In West Africa, Nigeria offers a potential market for *Megathyrsus maximus* hybrids of 33,938 ha. These figures imply that the mere participation of South Sudan, Ethiopia, and Tanzania represents about 67% of the potential area of adoption of *Megathyrsus maximus* hybrids in the analyzed countries, and 72% when only the East African countries are considered. These results show the most representative markets in the analyzed countries, according to areas best suited to adopt one of the two technologies of analysis.

Although no previous studies have delved into this type of analysis for tropical forages, geographic profiling techniques through environmental, climatic, and edaphic conditions have been implemented to evaluate other crops (see Section 2) [44,45,46]. The methodological approach used in this article is thus in line with these studies regarding two aspects, namely (i) the identification of potential areas to successfully implement a specific crop, and (ii) the identification of the production potential to be significantly higher than the current one, given the expected yields of the new technologies with superior characteristics.

### 5.3. Market Values

The estimation of the commercial value complements this analysis and is an essential element for decision-making by dairy producers, the private forage seed sector, and the public sector. The consultation of current market prices for both commercialized *Urochloa* cultivars and interspecific *Urochloa* hybrids resulted in a price premium of 37% for the hybrids. For *Megathyrsus maximus*, no hybrids are on the market yet that would serve for price comparison and guidance. To get an idea of the price of a higher quality material of *Megathyrsus maximus*, the *Urochloa* market was used as a reference and based on the differential found there, the potential price premium for hybrid seeds of *Megathyrsus maximus* was estimated at 37%. In a context where buyers are aware of the improved characteristics of new materials, willingness to pay is expected to be higher, since a greater investment will be compensated by efficiency gains, as well as higher productivity and income, as shown in a study on potato seed from Indonesia, where price increases in private sector markets of between 6% and 37% could be absorbed by the additional yields [38]. The estimated annual market values in millions of US$ are presented in Figure 4.

The total annual market value for both technologies is estimated at US$ 174,665,945, out of which *Megathrysus maximus* hybrids make up 58% and new interspecific *Urochloa* hybrids 42%. Regarding new interspecific *Urochloa* hybrids, the annual market value is US$ 73,521,066, and the largest market shares are held by Ethiopia, Tanzania, and Kenya with values close to US$ 19.8, 18.6, and 13.4 million, respectively. Regarding *Megathyrsus maximus* hybrids, the annual market value is US$ 101,144,879, and the largest market shares are held by South Sudan, Ethiopia, and Tanzania with values close to US$ 24.7, 22.1 and 20.8 million, respectively. Table 5 provides a summary on the potential market and annual market values for the analyzed countries.

On the other hand, it should be noted that the assumption of an adoption rate of 15% implies, at least theoretically, that the adoption of the new materials can occur in less than seven years, which means that the total market values for new interspecific *Urochloa* and *Megathyrsus maximus* hybrids would be US$ 490,140,439 and US$ 674,299,195, respectively. The obtained estimates for both the annual and total market values for the two hybrid forage materials are in line with the scarce literature and estimations on commercial values and growth potential of tropical forages in the global South. Private research companies such as Morder Intelligence valued the global forage market for 2020 at approximately US$ 20.33 billion and project that by 2026, it will reach about US$ 30.91 billion [86]. Calculations for Brazil indicate that, in 2019, the seed trade for tropical grasses exceeded 1.4 billion Reais, which was equivalent to almost US$ 269 million, noting that low-quality seeds participate with 30% of the total market [87]. A study on the extent and economic significance of cultivated forages in developing countries estimated the current total value of planted forages in developing countries at US$ 63 billion, corresponding to a coverage of 159 million hectares [23].

### 5.4. Requirements for a Development of This Market and Widespread Adoption of New Forage Hybrids

The results presented in this article suggest significant possibilities for both the commercialization of the new forage hybrid seeds and their adoption by dairy farmers. Growing improved forages as feed for dairy cattle is a valuable alternative to address the problems of food security and malnutrition in the region, since they increase both animal productivity and meat and milk quality. In this way, the change from production systems based on traditional pastures to systems that involve highly productive and more sustainable technologies would have positive effects on the poorest and most vulnerable population.

Another set of studies aims at estimating actual and potential adoption rates through impact evaluation techniques [40,41,42,43]. Although they are not focused on market segmentation by product, as is the objective of this research, they do contribute to the discussion by providing empirical evidence on the importance of adoption factors, such as the dissemination of knowledge and provision technical assistance to the potential users of the new technologies. The present study did not focus on the analysis of adoption factors but instead on providing decision-making support for the forages seed sector on potential opportunities for investment. However, investments in forage hybrid seed production and dissemination alone will not suffice to increase adoption rates of new technologies among dairy producers in East Africa despite tackling the lack of basic seed [30]. Literature, mostly for Latin American where the adoption of cultivated forages is more advanced compared to East Africa, shows that it depends on numerous additional factors. These include risk factors (risk aversion, perception of risks regarding future returns) [88,89,90], knowledge and information about the technology itself (establishment and management processes and costs, benefits and risks associated with the technology) [91,92,93], labor requirements [94], access to productive inputs and capital (credit) [95,96,97], product differentiation strategies [98], extension and technical assistance [47,92,99,100,101,102], the knowledge and innovation system [47,103,104], social capital and social networks (e.g., through farmer groups) [85,105,106,107,108,109], land prices, land tenure, land speculation [71,110,111], existing and evolving regulatory frameworks and political/institutional factors [112,113,114], and conflict [115].

Regarding regulatory frameworks and political and institutional factors, several studies, such as those by Enciso et al. [47], Orr [116], or Karandikar et al. [117], have shown that the institutional sector is important for facilitating or undermining the dissemination and adoption of improved agricultural technologies such as forage hybrids. Public policies without a clear focus can create distortions in the process. Technological developments and the marketing of new hybrid forages need to integrate the private sector with public extension, research, and distribution systems [47,117]. The confluence and cooperation of actors allow structuring policies according to the local realities and needs of the targeted producers. In short, the sustainability of these systems depends on the collaboration of different actors, which is not guaranteed in East Africa as studies from Kenya, Tanzania, and Ethiopia show [31,32,33]. Likewise, the regulatory frameworks for forage seed production and certification in East Africa are complex and in cases too demanding for seed companies to comply with, leading to withdrawals from both seed bulking and production, for example in Kenya [31] and Tanzania [32], and to weak formal seed systems. Informal seed systems, for example the exchange of seeds or vegetative material among smallholder farmers, on the other hand, are often marginalized, incriminated, and not considered in the legal frameworks in the region [118,119].

Under the umbrella of the so-called Industrial Revolution 4.0 (IR4.0), digital technologies such as the Internet of Things, Big Data, and Artificial Intelligence have been increasingly incorporated in food systems [120]. Regarding the topic of the present study, the IR4.0 offers numerous opportunities for both the development of a hybrid forage seed sector in East Africa and the adoption of the hybrids by dairy farmers. Big data, for example, are already being used to support the selection of tropical forages based on specific agro-ecological conditions with clear indications on how to grow the materials, and thus reduce the risk of failure in forage adoption [121]. Likewise, mobile applications are being developed that include Artificial Intelligence to support dairy farmers with decision-making, such as the DigiCow and Digital Dairy apps in Kenya [122,123].

In summary, both the results of the present study as well as the above-described evidence on the possibilities and limitations for the adoption and dissemination of new forage hybrids in East Africa can be related to the Quintuple Helix innovation model [124]. According to the model, socio-ecological transition can only happen if collaboration among a broad set of actors happens, i.e., among the higher education, economic, and political systems, and if this is put into the context of both a media- and culture-based public and the natural environment of society. For the case of forage hybrid adoption for increasing food security and environmental sustainability in East African dairy systems, collaboration among the different actors is thus essential. This includes national and international research and education institutions, forage seed producers and distributors, forage and dairy value chain actors, public sector actors for seed regulation and extension, and financing institutions. It also includes the consideration of social capital (e.g., traditions and values) and capital of information (e.g., communication, social networks), as well as natural capital (e.g., resources, environmental conditions) in both the development and scaling of new forage technologies.

Finally, it is worth mentioning that most market research on improved crops focuses on identifying and delimiting the geographical areas with the greatest possibilities for adoption. Likewise, market studies on improved forages are scarce. The present article combined two methodological approaches, namely (i) the estimation of potential areas where two hybrid forages can be planted and the amount that is needed by the dairy cattle herd present in the countries of analysis, considering specific environmental and productive conditions present in the region of analysis, and (ii) the estimation of the commercial value for each of the two technologies in each of the analyzed countries. Both the methodological approach and obtained results presented above thus add significant value to the scientific readings on the subject.

## 6. Conclusions

The estimations of potential markets (market sizes and values) for new interspecific *Urochloa* and *Megathyrsus maximus* hybrid forages for the dairy sector in East Africa provided in this article indicate an important opportunity for making changes in the local food systems. Moving from dairy systems based on traditional or low-quality pastures towards systems that integrate improved forage materials, i.e., hybrids, provides opportunities for improving both the availability and nutritional quality of animal source food, i.e., milk, and thus contributes to achieving food security and combating hunger in the region. Likewise, taking advantage of and developing these markets implies an opportunity to promote a sector that has the capacity to generate income and livelihoods for the most vulnerable part of the population. From a point of view of economic development, the promotion and consolidation of these markets can be an effective economic policy to improve indicators of poverty, unemployment, growth, and price stability, since dairy is a fundamental activity for the economic structure in the region and its development has a positive impact on the entire macroeconomic environment. The development of potential forage hybrid markets, however, requires that adequate market conditions exist. In this sense, a favorable commercial and institutional environment needs to emerge that supports the production, distribution, and adoption of forage hybrids. This is a determining element since it will provide a regulatory environment that attracts investments in seed production and distribution as well as the necessary incentives for dairy producers (e.g., access to knowledge, seeds and other inputs, or credit) to make informed adoption decisions. Alongside, communication and collaboration between the various actors must be enhanced to jointly work on the difficulties that arise with the adoption of forage hybrids. An adequate information system is essential for decision makers to establish policies and implement timely actions according to the local contexts. Considering these aspects is thus essential for the development of a competitive forage hybrid seed market so that promising materials can be properly registered and made available to farmers.

## Figures and Tables

**Figure 1 foods-12-01607-f001:**
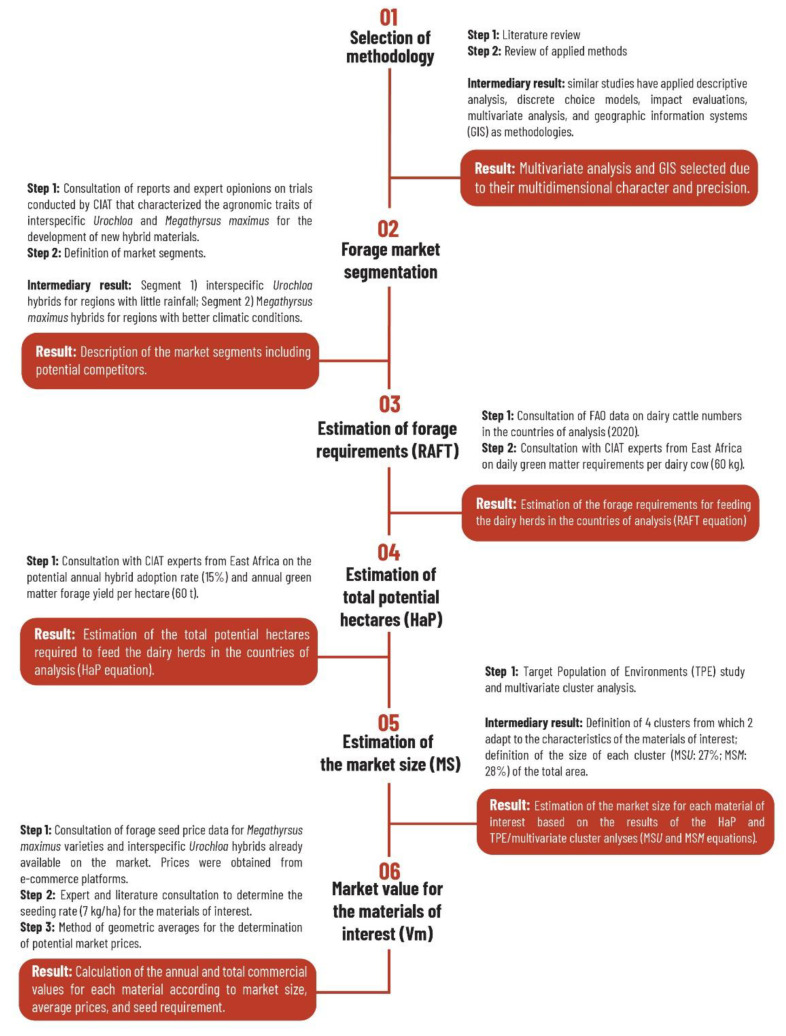
Methodological flowchart for the estimation of potential markets of hybrid forages in Africa.

**Figure 2 foods-12-01607-f002:**
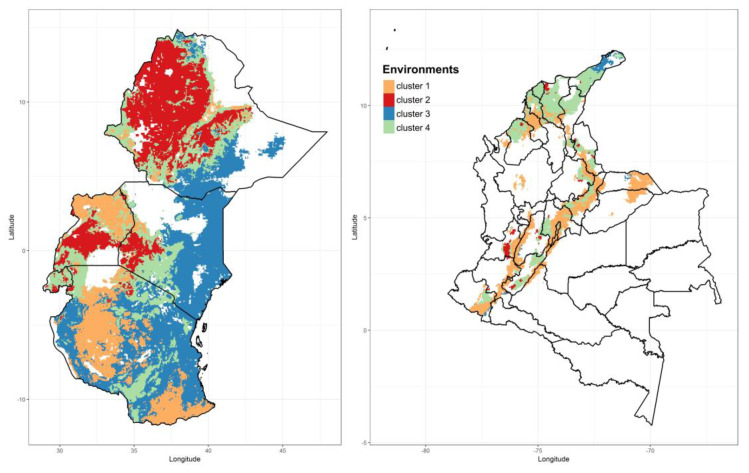
(**left**) Identified geographic clusters for East Africa and (**right**) Colombia [37].

**Figure 3 foods-12-01607-f003:**
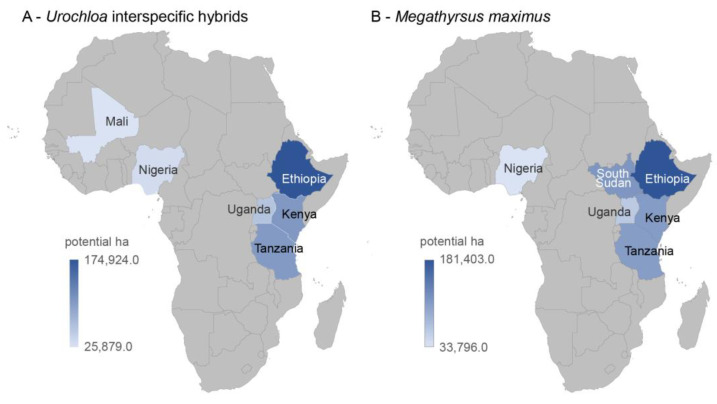
Market size for new interspecific *Urochloa* (**A**) and *Megathyrsus maximus* (**B**) hybrids. Sources: [85] and (V. Castiblanco and A. Notenbaert personal communication, 13 June 2019).

**Figure 4 foods-12-01607-f004:**
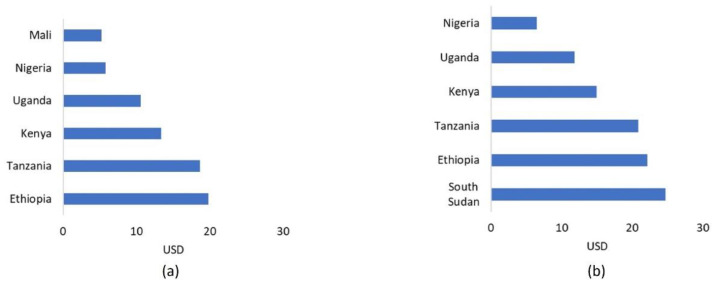
Estimated annual market values in millions of dollars for new interspecific *Urochloa* (**a**) and *Megathyrsus maximus* (**b**) hybrids.

**Table 2 foods-12-01607-t002:** Potential competitors for new *Megathyrsus maximus* hybrids.

Characteristics	Mombasa	Tanzania	Massai	Mavuno *
Main features	High regrowth rate and good stem-leaf-ratio Medium tolerance to cold and burning Good drought tolerance [72]	Medium drought tolerance [72,75]	Burn and shade tolerance Reduced yield by 50% in dry season [2,72]	Good tolerance to drought, burning, and shade Medium tolerance to humidity [61,76]
Resistance to pests and diseases	spittlebug [72]	spittlebug medium tolerance to coal in the inflorescences [72]	spittlebug sensitive to panicle rot caused by *T. ayresii* [72]	spittlebug [77]
Required soil fertility level	medium to high acid soils [72]	medium to high acid soils [72]	low to medium acid soils [72]	medium acid soils [61]
Palatability	very good [72]	good [75]	good [75]	very good [77]
CP (%)	10–14 [72]	10–12 [72]	7–11 [72]	18–21 [76]
IVDMD (%)	60–65 [72]	62 [72]	55–60 [72]	60 [76]
Yield (t/ha/cut)	25 [72]	18–20 [72]	21 [72]	17–20 [76]
Main use	grazing cut-and-carry [72]	grazing cut-and-carry [72]	grazing cut-and-carry [75]	grazing cut-and-carry [76]

Source: Own elaboration based on [2,61,72,75,76,77]. * Mavuno was released by Wolf Sementes from Brazil in 2013 [61]. Despite being an *Urochloa* hybrid, due to its high performance, it is considered a potential competitor in the *Megathyrsus maximus* market.

**Table 3 foods-12-01607-t003:** Variables and information sources.

Variable	Description	Source
Dairy cattle herd (CRL)	Dairy cattle heads per country in 2020	[78]
Daily forage requirement (RDFK)	Daily forage requirement in kg	Expert consultation, [2]
Adoption rate (AR)	Estimated adoption rate of new hybrids in %	Expert consultation, [2]
Average hybrid forage yield (Ytha)	Average hybrid forage yield per hectare in tons	Expert consultation, [2]
Proportion of potential area for new interspecific *Urochloa* hybrids	Proportion of potential area for new interspecific *Urochloa* hybrids in % defined using the TPE study	[37]
Proportion of potential area for *Megathyrsus maximus* hybrids	Proportion of potential area for *Megathyrsus maximus* in % defined using the TPE study	[37]
Sowing rate (S)	Sowing rate of forage hybrids in kg per hectare	Expert consultation, [79]
Seed price forage hybrids (P)	Average market price for a kg of hybrid seed	Expert consultation, [59,79]

**Table 4 foods-12-01607-t004:** Annual forage requirement by the dairy cattle herd and potential annual area for forages.

Country	Dairy Cattle Herd in 2020 (Heads) [78]	RAFT: Forage Requirement (Mt/y)	HaP: Potential Forage Area (ha/y)
Ethiopia	7,556,402	165,485,204	413,713
Tanzania	7,116,771	155,857,285	389,643
Kenya	5,112,340	111,960,246	279,901
Uganda	4,037,038	88,411,132	221,028
South Sudan	8,432,559	184,673,042	461,683
Nigeria	2,213,856	48,483,446	121,209
Mali	1,995,914	43,710,517	109,276

**Table 5 foods-12-01607-t005:** Summary of potential market sizes and values for new forage hybrids in Africa.

Country	New Interspecific *Urochloa* Hybrids		*Megathyrsus maximus* Hybrids		Vm: Total Annual Market Value (US$/Country)
	MS: Potential Market Size (ha)	Vm: Annual Market Value (US$)	MS: Potential Market Size (ha)	Vm: Annual Market Value (US$)	
Ethiopia	111,703	19,818,364	115,840	22,173,319	41,991,682
Tanzania	105,204	18,665,332	109,100	20,883,276	39,548,609
Kenya	75,573	13,408,261	78,372	15,001,524	28,409,785
Uganda	59,678	10,588,040	61,888	11,846,184	22,434,224
South Sudan	n/a	n/a	129,271	24,744,292	24,744,292
Nigeria	32,726	5,806,335	33,938	6,496,284	12,302,619
Mali	29,505	5,234,733	n/a	n/a	5,234,733
Total	414,388	73,521,066	528,409	101,144,879	174,665,945

## Data Availability

The data used were obtained from the FAO’s FAOSTAT statistical information platform. This information is freely accessible, https://www.fao.org/faostat/en/#data/QCL (accessed on 4 January 2023).

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
