# Peer review of "Potential Forage Hybrid Markets for Enhancing Sustainability and Food Security in East Africa"

_foods, 2023, doi:10.3390/foods12081607_

Round 1

Reviewer 1 Report

Very important matter.

However:

1. I suggest putting some introduction, saying what will be treated in this item, between items 2 and 2.1.

2. Item 2.1 is difficult to follow and understand, perhaps a figure illustrating the procedure should be included.

3. I suggest joining items 3. and 4.

4. Item 5. Conclusions is too verbose. Some things presented there cannot be supported by what the work shows. I suggest that what cannot be fully supported by the results of the work be moved to items 3 and 4.

Author Response

Dear reviewer, thanks for the detailed and good review. We have incorporated your suggestions and are sure to have addressed them adequately. They have helped to enriching our article and improving its quality. Don't hesitate in contacting us for further clarifications.

Reviewer 2 Report

Dear Author

Thank you for submitting in Foods. The designed study has been good but presentation of the work does not meet the publication level. I am confused to consider it article, case report etc. It should be at least clear. English Editing is required.

The title is very much broad and visionary, but the data compaction is not up to the mark. So major comments are below

The introduction has been observed somewhat far from the actual theme of the manuscript. A story form is at least required for the background text of title and the goal of the study. 

Citations are much needed to improve. 

No objectives of the study has been mentioned.

In materials and methods

"Methodological background and brief literature review on similar studies", What is the purpose of this here? is this survey based study?

Other sections are also too extensive and does not reveal objectives/goal.

Statistical analysis is missing as well.

Results are also very weak and had no presentation. 

Discussion is hard to understand and read in the presence of current materials and methods and results.

Line 29-41. The provided citation does not see so valid. I will suggest to cite the original article/Agency or provide link of the source to find this.

Author Response

(The authors gave the same response as above.)

Reviewer 3 Report

The manuscript describes the potential forage hybrid markets for enhancing sustainability and food security in East Africa. The manuscript need amendments prior to publication.

1. In abstract - short introduction too long, methodology not clear.
2. Introduction
- line 45/46 no reference?
- no clear objectives in the last paragraph
3. Provide overall flowchart of methodology, so that reader could understand better. Methodology not clear. how many samples? secondary data? year involve? social science method? statistical analysis?
4. Table 1 and 2 not clear. what do you mean by own elaboration? which data is referring to reference 35, 37, 38, 48-50? 37, 49, 52–54? 
5. Two tables 1?
6. Provide references for all equations used.
7. Methodology provided too lengthy. Should be specific.
8. Results and discussion should be combined. I have major concern here. The results were insufficient for Q1 publication.  only 2 figures?
9. Discussion - how about the national/international policy/governance on forage hybrid markets?
10. How about the IR4.0 in food security? Quintuple helix framework? Need to add on discussion.
11. Conclusion too long. Make it one paragraph to answer objectives
12. What is the contribution/novelty of the research?

Author Response

(The authors gave the same response as above.)

Reviewer 4 Report

This is an interesting paper. However, some issues should be corrected.

1. Citation should be corrected.

2. Please provide research questions.

3. Write the research hypothesis.

4. Write how this paper is organised at the end of introduction.

5. Explain all abbreviation and prepare nomenclature at the end of the paper.

Author Response

(The authors gave the same response as above.)

Round 2

Reviewer 1 Report

The authors have improved the text.

Author Response

Thank you

Reviewer 2 Report

Dear Authors

Thank you for you revised version. Many changes has been done by the authors. Article is in better form now. Citations numbers are misplaced throughout the manuscript and correction should be incorporated.

Author Response

Thanks. The references were organized correctly and the citation numbers adjusted.  

Reviewer 3 Report

Still need amendments:

Introduction 
- Line 166-173 (this is not objectives)
- Provide clear objectives

1. Introduction

- no clear objectives in the last paragraph

Author Response

The hypothesis, research questions, and objectives were clearly outlined in the fifth paragraph of the introduction (Lines 83-100 without track changes, or Lines 157-178 with track changes). Please see here what was included:

"Under this scenario, the hypothesis of this research article is that an important potential market exists for new forage hybrids in East Africa that can be captured by the private forage seed sector and contribute to increasing livelihoods, food security, and nutrition in the region. This market basically emerges from the huge potential for adoption of hybrid forages resulting from the superior characteristics the materials offer regarding productivity, adaptability to the environment, and nutritional quality that improve both animal productivity and livelihoods of dairy farmers, while contributing to food security and environmental sustainability in the region. The mentioned hypothesis leads to the following research questions: (i) How big is this potential market for new forage hybrids of Urochloa and Megathyrsus maximus in East Africa, and (ii) what needs to be done so that the potential can be captured and adoption will happen? The objective of this article is thus to provide an analysis of this potential market for new interspecific Urochloa and Megathyrsus maximus hybrids for East Africa, and some West African countries. Particularly, this study aims at (i) providing a market segmentation for the two hybrids of interest, (ii) estimating the area that can be covered by the two hybrids in each country of analysis (size of potential market), (iii) estimating the commercial annual and total values the two hybrids could generate in the region (market value), and (iv) describe how this market can be captured to make large-scale adoption reality."
